# The Platelet-Specific Gene Signature in the Immunoglobulin G4-Related Disease Transcriptome

**DOI:** 10.3390/medicina61010162

**Published:** 2025-01-19

**Authors:** Ali Kemal Oguz, Cagdas Sahap Oygur, Bala Gur Dedeoglu, Irem Dogan Turacli, Sibel Serin Kilicoglu, Ihsan Ergun

**Affiliations:** 1Department of Internal Medicine, Faculty of Medicine, Ufuk University, 06510 Ankara, Turkey; 2Department of Internal Medicine (Rheumatology), Faculty of Medicine, Baskent University, 06490 Ankara, Turkey; cagdasahap@yahoo.com; 3Department of Biotechnology, Biotechnology Institute, Ankara University, 06135 Ankara, Turkey; gurbala@yahoo.com; 4Department of Medical Biology, Faculty of Medicine, Ufuk University, 06510 Ankara, Turkey; doganirem@gmail.com; 5Department of Histology & Embryology, Faculty of Medicine, Baskent University, 06790 Ankara, Turkey; sibelserin@yahoo.com; 6Department of Internal Medicine (Nephrology), Faculty of Medicine, Ufuk University, 06510 Ankara, Turkey; ihsanerg@yahoo.com

**Keywords:** blood platelets, immunoglobulin G4-related disease, gene expression profiling, transcriptome, platelet activation, fibrosis

## Abstract

*Background and Objectives:* Immunoglobulin G4-related disease (IgG4-RD) is an immune-mediated, fibroinflammatory, multiorgan disease with an obscure pathogenesis. Findings indicating excessive platelet activation have been reported in systemic sclerosis, which is another autoimmune, multisystemic fibrotic disorder. The immune-mediated, inflammatory, and fibrosing intersections of IgG4-RD and systemic sclerosis raised a question about platelets’ role in IgG4-RD. *Materials and Methods*: By borrowing transcriptomic data from Nakajima et al. (GEO repository, GSE66465) we sought a platelet contribution to the pathogenesis of IgG4-RD. GEO2R and BRB-ArrayTools were used for class comparisons, and WebGestalt for functional enrichment analysis. During the selection of differentially expressed genes (DEGs), the translationally active but significantly low amount of platelet mRNA was specifically considered. The platelet-specific gene signature derived was used for cluster analysis of patient and control groups. *Results*: When IgG4-RD patients were compared with controls, 268 DEGs (204 with increased and 64 with decreased expression) were detected. Among these, a molecular signature of 22 platelet-specific genes harbored genes important for leukocyte–platelet aggregate formation (i.e., *CLEC1B*, *GP1BA*, *ITGA2B*, *ITGB3*, *SELP*, and *TREML1*) and extracellular matrix synthesis (i.e., *CLU*, *PF4*, *PPBP*, *SPARC*, and *THBS1*). Functional enrichment analysis documented significantly enriched terms related to platelets, including but not limited to “platelet reactivity”, “platelet degranulation”, “platelet aggregation”, and “platelet activation”. During clustering, the 22 gene signatures successfully discriminated IgG4-RD and the control and the IgG4-RD before and after treatment groups. *Conclusions*: Patients with IgG4-RD apparently display an activated platelet phenotype with a potential contribution to disease immunopathogenesis. If the platelets’ role is validated through further carefully designed research, the therapeutic potentials of selected conventional and/or novel antiplatelet agents remain to be evaluated in patients with IgG4-RD. Transcriptomics and/or proteomics research with platelets should take into account the relatively low amounts of platelet mRNA, miRNA, and protein. Secondary analysis of omics data sets has great potential to reveal new and valuable information.

## 1. Introduction

The accumulation of evidence has led us to recognize platelets, the key player of hemostasis and maintenance of vascular integrity, also as immune cells with regulatory and effector functions [1,2,3]. These anucleated cells of the myeloid lineage appear to perform these immune functions mainly through their surface proteins and bioactive molecules stored and secreted from their cytoplasmic granules, and also by directly interacting with various leukocytes [4,5]. Another important and closely related function of platelets appears in the role they play in tissue remodeling, which characteristically occurs after an injury or an inflammatory process [6,7]. At this point, systemic sclerosis would be a demonstrative example with its immune-mediated, multisystemic fibrotic and aberrant platelet activation characteristics [8,9]. Demonstration of the role of platelets in pathological immune processes and elucidation of the molecular mechanisms responsible for platelet contribution to these disorders may enable the scientific community to develop novel targeted therapies for these conditions.

Immunoglobulin G4-related disease (IgG4-RD) is a relatively new disease entity that is an inflammatory disorder characterized by tissue infiltration of IgG4(+) plasma cells, accompanying storiform fibrosis of the affected tissues, and obliterative phlebitis [10,11,12]. A recent review reports that IgG4-RD tends to occur during the fifth to seventh decades of life, although both pediatric and elderly patients are also observed [13]. With regard to gender, while most IgG4-RD studies document a male predominance, this gender difference shows variation with the tissues/organs involved (i.e., pancreatobiliary and retroperitoneal involvement is more common among males and disease confined to the head and neck region is more common among females) [13]. Although population-specific measures of incidence and prevalence are scarce in the literature, the estimated incidence was reported to be around 0.78–1.39 per 100,000 person-years in the USA [14]. This multisystemic fibroinflammatory condition which is increasingly being recognized exhibits common features with systemic sclerosis, such as being an immune-mediated condition, harboring inflammatory findings, and displaying tissue fibrosing characteristics [15]. Although the immunological aberrations in IgG4-RD have been related to the innate and adaptive arms of the immune system, the disorder has not yet been defined exactly in terms of etiology and pathogenesis [16,17]. In addition, findings that point to a potential role of platelets in the pathogenesis of IgG4-RD are extremely scarce in the relevant literature [18]. Nevertheless, significant findings pointing to a strong platelet activation and contribution to disease pathogenesis in systemic sclerosis and systemic sclerosis’s and IgG4-RD’s shared fibroinflammatory features raise a question about platelets’ contribution to IgG4-RD immunopathogenesis.

In our present study, by borrowing the microarray expression data from the study by Nakajima et al. (Gene Expression Omnibus data repository [GEO], accession number GSE66465) and implementing a secondary data analysis approach, we sought to document any potential contribution of platelets to IgG4-RD pathogenesis [19]. A purposively chosen fold change (FC) cut-off, taking into account the translationally active, albeit significantly small amount of platelet mRNA, and a detailed functional enrichment analysis from various functional databases were our study’s two distinctive bioinformatics approaches. Even though this was a preliminary in silico analysis, the findings were remarkable, as they pointed to a potential contribution of platelets and platelet activation in the pathogenesis of IgG4-RD.

## 2. Materials and Methods

As the study involved an “in silico” secondary analysis of a publicly available transcriptomic data set on IgG4-RD, ethical approval was not sought. A flow diagram of the study is presented in Figure 1.

### 2.1. The Gene Expression Profiling Study by Nakajima et al.

Two patients with IgG4-RD diagnosed according to the comprehensive diagnostic criteria and 4 healthy control subjects were enrolled in the gene expression profiling study by Nakajima et al. (GEO accession GSE66465) [19,20]. The comprehensive diagnostic criteria for IgG4-RD (2011) by Umehara et al. involved two diagnostic criteria: (1) serum IgG4 concentration >135 mg/dL and (2) >40% of IgG(+) plasma cells being IgG4(+) and >10 cells/high-power field of biopsy sample [20]. The two IgG4-RD patients were males with the ages of 66 (patient 1) and 63 (patient 2) [19]. The four control subjects were also all males, with ages ranging from 57 to 64 [19]. The tissue/organ involvement of the two patients were the salivary gland/duodenum/lymph node and salivary gland/bile duct/pancreas/prostate for patient 1 and patient 2, respectively [19]. The patients were evaluated before and after 3 months of steroid therapy (0.6 mg/kg/day with dose reduction of 10% every 2 weeks) [19]. The total RNA was isolated from peripheral blood mononuclear cells and GeneChip^®^ Human Gene 1.0 ST Arrays (Affymetrix^®^, Santa Clara, CA, USA) were used for hybridization [19]. The details regarding the RNA isolation and the microarray hybridization steps are presented in Nakajima et al.’s paper [19]. Raw and processed data were deposited in the GEO data repository with the series ID GSE66465.

### 2.2. Retrieval and Class Comparison Analysis of Gene Expression Data

The microarray data of the study by Nakajima et al. were retrieved from GEO by using the GEO accession number GSE66465 on 9 September 2022 [21,22]. Class comparison analyses between the IgG4-RD patients and control groups were performed using GEO2R [23]. As stated on the GEO website, “GEO2R is an interactive web tool that allows users to compare two or more groups of samples in a GEO series in order to identify genes that are differentially expressed across experimental conditions”. The default settings of GEO2R were preserved (i.e., “Benjamini–Hochberg procedure” to control the false discovery rate, “auto-detect” option to apply log transformation to the data, “no” for the vooma function, “no” for forcing normalization, “NCBI generated” annotations and “0.05” for the cut-off level of significance). Successively, (1) treatment-naive patients with IgG4-RD (before treatment [BT]) versus healthy controls (HC) and (2) BT versus glucocorticoid-treated patients with IgG4-RD (after treatment [AT]) class comparisons were performed (BT vs. HC and BT vs. AT, respectively). Ultimately, GEO2R generated two spreadsheets of differentially expressed gene (DEG) sets. While the first one of these two files listed the top 250 DEGs sorted according to their *p* values, the second one included all the microarray/probe set data, again sorted according to the *p* values.

### 2.3. Selection of DEGs for Downstream Analysis and Functional Enrichment Analysis

Differentially expressed genes for subsequent analysis were selected using an FC criteria |log_2_FC| ≥ 1 (as FC values are log base 2 transformed, absolute values of ≥1 mean changes in expression levels of at least two times in increasing or decreasing directions). Additionally, FC values of ≥1.5, ≥2, ≥3, ≥4, and ≥5 were also implemented to construct the respective DEG sets.

Functional enrichment analysis of DEGs was performed with the WEB-based Gene Set Analysis Toolkit (WebGestalt) [24,25]. By implementing an over-representation analysis (ORA) method, functional database categories of gene ontology (sub-root of biological process), pathway (Panther), and disease (GLAD4U) were searched. Again, the default settings were preserved for the advanced parameters of WebGestalt. During the choice of DEG sets for functional enrichment analysis, different FC cut-off values were used (i.e., ≥1, ≥1.5, ≥2, and ≥3). Taking into account the relatively low amount of platelet mRNA content, which is estimated to be around 1/12,500 of the leukocytes’, an absolute FC value of ≥1 was chosen as the starting point.

### 2.4. Venn Diagram and Cluster Analysis

The Venn diagram analysis of the DEGs of the study and the Reactome Pathway Database’s gene set (Platelet activation, signaling and aggregation, R-HSA-76002) was performed with Venny 2.1.0 [26,27,28].

For the cluster analysis of IgG4-RD patient groups and HC subjects, the built-in clustering tools of the BRB-ArrayTools software (v4.6.2, developed by Dr. Richard Simon and the BRB-ArrayTools Development Team) and the Java TreeView (v1.2.0) and Cluster 3.0 software packages were used [29,30,31,32]. During clustering, both the samples and the genes were clustered and an unsupervised hierarchical clustering algorithm using a Euclidean distance metric and an average linkage was implemented. The set of genes used for clustering was generated from the platelet-specific molecular signature made up of 22 genes.

## 3. Results

### 3.1. Class Comparison Analysis

The results of the class comparison analysis are summarized in Table 1. The complete DEG lists with details are presented in Appendix A.

The top 30 DEGs (15 with increased and 15 with decreased expression levels), sorted according to their FC values, are presented in Table 2. Interestingly, while the functional enrichment analysis documented many significantly enriched terms related to platelets and platelet activation, the top 30 DEG list harbored mainly leukocyte-related innate immune response genes. When the FC values of the top 30 DEGs and the significantly lower amount of platelet mRNA with respect to leukocytes’ (platelet/leukocyte: 1/12,500) are taken into account, this finding has a good explanation.

The findings of the class comparison analysis fundamentally showed that, while an FC cut-off ≥2 mainly revealed innate immunity genes, a cut-off value of ≥1 exposed platelet and platelet activation-related genes.

### 3.2. Functional Enrichment Analysis

The results of the functional enrichment analysis are graphically represented in Figure 2. Again, comprehensive results of the functional enrichment analysis with related details are given in Appendix A. Strikingly, the enriched terms related to platelets and platelet function unfolded only when the FC cut-off value was selected as ≥1 (Appendix A). At this point, it is important to remember that the FC values are log base 2 transformed and an absolute FC value of ≥1 would mean at least a twofold change in expression level in either direction (i.e., increasing or decreasing) between the compared classes.

As seen in Figure 2, many terms related to platelets were significantly enriched in the functional databases of the gene ontology, pathway, and disease. The enrichment ratios (ER) of these terms were found to be between 6.4 (“arterial occlusive diseases”) and 83.2 (“thrombasthenia”) (Appendix A). With their ER values in decreasing order, “thrombasthenia” (ER: 83.2), “low on-treatment platelet reactivity” (ER: 55.9), “high on-treatment platelet reactivity” (ER: 43.4), “platelet reactivity” (ER: 40.8), “platelet aggregation inhibition” (ER: 23.5), “blood platelet disorders” (ER: 21.7), “platelet degranulation” (ER: 20.2), “platelet aggregation” (ER: 19.3) and “platelet activation” (ER: 18.3) were significantly enriched platelet-specific terms, drawing considerable attention during functional enrichment analysis.

The list of shared genes appearing both among our DEGs and also among the constituting genes of the platelet-specific terms mentioned above is presented in Table 3. Table 3 also briefly summarizes key platelet-specific functions of these 22 genes (that is, in alphabetical order, *ALOX12*, *CLEC1B*, *CLU*, *CMTM5*, *GP1BA*, *ITGA2B*, *ITGB3*, *MIR223*, *MMRN1*, *MPL*, *P2RY12*, *PDE5A*, *PF4*, *PF4V1*, *PPBP*, *PROS1*, *PTGS1*, *SELP*, *SPARC*, *THBS1*, *TREML1*, and *TUBB1*). For the subsequent cluster analysis, this gene list was used as a platelet-specific molecular signature. It is worth mentioning that all transcripts had increased expression levels in this gene list except *MIR223*.

If the main findings of the functional enrichment analysis are to be summarized, (1) the significant enrichment of platelet and platelet activation-related terms and (2) the derivation of a 22-gene transcriptomic platelet-specific signature indicating platelet activation and contribution in IgG4-RD should particularly be mentioned.

### 3.3. Venn Diagram and Cluster Analysis

To further support the platelet association of our derived 22-gene molecular signature, Reactome Pathway Database’s “platelet activation, signaling and aggregation” (R-HSA-76002) pathway’s gene list was taken as a reference. The Venn diagram analysis of the platelet-specific gene signature and the Reactome Pathway Database-derived gene list is presented in Figure 3. Fourteen of the 22 constituent genes of the gene signature were present at the intersection “REACTOME ∩ DEGs PLATELET” (i.e., *CLEC1B*, *CLU*, *GP1BA*, *ITGA2B*, *ITGB3*, *MMRN1*, *MPL*, *P2RY12*, *PF4*, *PPBP*, *PROS1*, *SELP*, *SPARC*, and *THBS1*). Notably, 5 genes with activation and leukocyte–platelet aggregate formation functions (i.e., *CLEC1B*, *GP1BA*, *ITGA2B*, *ITGB3*, and *SELP*) and 5 others with important extracellular matrix deposition functions (i.e., *CLU*, *PF4*, *PPBP*, *SPARC*, and *THBS1*) were among these 14 genes. The complete gene list of the Reactome Pathway Database’s R-HSA-76002 is given in Appendix A.

The clustering dendrograms and heat maps of the BT samples with the HC and AT samples are presented in Figure 4 (panels (A) and (B), respectively). As seen in Figure 4, the platelet-specific gene signature and the clustering algorithm implemented were successful in clustering the BT, HC, and AT samples.

As a selected gene with important platelet activation and degranulation functions, the graphical representations of the relative expression levels of the *SELP* gene are also shown in Figure 4 (panel (C) for BT vs. HC and panel (D) for BT vs. AT).

Strikingly, a 22-gene platelet-specific molecular signature effectively clustered before-treatment IgG4-RD patients, after-treatment IgG4-RD patients, and healthy controls in an error-free manner.

### 3.4. The Results of Nakajima et al. Compared [19]

As previously stated, a purposively chosen FC cut-off and a comprehensive functional enrichment analysis from various functional databases were our two distinctive approaches compared to Nakajima et al.’s study. The study by Nakajima et al. performed an in vitro microarray-based transcriptomic analysis and included BT vs. HC and BT vs. AT class comparisons, and a real-time PCR validation of selected hypothetically important genes [19]. While Nakajima et al. showed the decreased expression of certain allergy- or innate immunity-related genes in patients with IgG4-RD, they did not choose to perform functional enrichment or cluster analysis with their DEG sets [19]. In this way, the results pertaining to platelets and platelet activation, specifically including the functional enrichment analysis findings, the 22-gene molecular signature derived, and the cluster analysis findings presented above, are the present study’s distinctive findings.

## 4. Discussion

Platelets, with their active immune regulatory and effector functions and significant tissue remodeling capabilities, have great potential to contribute to the pathogenesis of fibroinflammatory conditions. This has been proven to be true in certain inflammatory disorders with fibrosing features, including but not limited to systemic sclerosis, cystic fibrosis, chronic hepatitis, idiopathic pulmonary fibrosis, and cardiac inflammation and fibrosis [8,33,34,35,36]. The present study, by performing a secondary data analysis of an IgG4-RD transcriptomic data set, documented significant platelet-specific functional enrichment terms and a molecular signature indicative of platelet activation, which strongly support platelet contribution to the pathogenesis of IgG4-RD.

In their original research article, Nakajima et al. aimed to investigate the pathogenesis of IgG4-RD by performing a comparative transcriptomic analysis and chose an FC value of ≥3 to identify their DEGs [19]. According to the findings of their study, Nakajima et al. concluded that genes related to innate and allergic immune responses (i.e., *CLC*, *MS4A3*, *DEFA3*, *DEFA4*, *IL8RA*, and *IL8RB*) were negatively regulated in patients with IgG4-RD and this altered expression pattern could have contributed to the disease pathogenesis. The original work of Nakajima et al. did not include a functional enrichment analysis of their DEGs [19]. Additionally, as the authors, we believe that the FC cut-off chosen by Nakajima et al. (i.e., ≥3) might have compromised the documentation of platelets’ transcriptomic contribution to IgG4-RD pathogenesis.

Once thought of as being purely cellular fragments with restricted hemostatic function, platelets are now at the forefront of immunology with the spectacular roles they play during innate and adaptive immune responses [37]. Today, it is also clearly documented that megakaryocytes, the bone marrow progenitors of platelets, selectively and actively sort mRNAs to the newly formed platelets during thrombopoiesis [38,39]. Although the platelet transcriptome contains a considerably low amount of mRNA, it has been shown to be dynamic (i.e., adapting to and reflecting distinctive physiological and pathological stimuli), translationally active, and closely reflecting the platelet proteome [39,40,41,42,43,44]. Regarding platelet mRNA content, it is estimated that leukocyte mRNA content is 12,500 times higher than that in the platelet [45]. This significantly small amount of platelet mRNA was the main reason for the chosen FC cut-off value of ≥1 in our study (Table 1).

As mentioned above, platelets, the ubiquitous component of blood, are increasingly recognized for their immune effector and modulatory functions [1,3,46]. From a phylogenetic perspective, it will be significant to remember that invertebrates’ hemocytes, the ancestors of platelets, are equipped with both phagocytic and hemolymph coagulating functions for defensive purposes [2,47]. Although most of the current data are about the interplay of platelets and innate immunity, there is an exponentially growing body of evidence regarding the interaction of platelets and adaptive immunity as well [48,49]. While a detailed review is beyond the scope of this article, contribution to the formation of inflammatory edema, recognition and sequestration of pathogens, recruitment and activation of leukocytes, triggering of inflammasome activation, enhancement of the diverse killing functions of phagocytes, antigen presentation through MHC class I molecules, and response to pathogens via their surface FcγRIIA in already immune hosts are among the reported immune functions of platelets [2,50,51,52,53]. The research findings demonstrate that platelets perform these functions through their (1) cell surface receptors, (2) soluble mediators, including cytokines and chemokines stored and secreted from their dense and α-granules, and (3) shed microparticles [5,54,55,56]. In addition to their secreted products, platelets generate neutrophil–platelet, monocyte–platelet, and lymphocyte–platelet aggregates, which help platelets communicate and interact directly with leukocytes [57]. These leukocyte–platelet aggregates are known to increase substantially during inflammatory disorders and allow both cellular components of the aggregate (i.e., leukocyte and platelets) to reciprocally exchange molecules and cytoplasmic constituents, with potential phenotypic alterations in both cells [57,58].

Our platelet-specific gene signature contained many genes indicative of immune-mediated platelet activation (Table 3). Among these, *CLEC1B* (C-type lectin domain family 1 member B) is a powerful innate immune receptor that activates platelets and functions through an alternative (“nonclassical”) platelet activation pathway [59]. The genes *GP1BA* (glycoprotein Ib platelet subunit alpha), *ITGA2B* (integrin subunit alpha 2b), *ITGB3* (integrin subunit beta 3), *SELP* (selectin P), and *TREML1* (triggering receptor expressed on myeloid cells like 1) are all molecules of the cell surface and have been shown to play important roles in the formation of various leukocyte–platelet aggregates [57,58,60,61].

One of the hallmarks of IgG4-RD is the “storiform” fibrosis of the affected tissues and organs [12]. The molecular mechanisms underlying this profound and characteristic fibrosis are not yet fully clarified, but profibrotic stimuli provided by infiltrating B cells, T cells (CD4+ T cells with cytotoxic activity, T follicular helper cells), and activated macrophages (M2 macrophages) are held responsible in the literature [62,63,64]. However, based on the findings of our study, we believe that it is critical to remember platelets as one of the key players in tissue repair, regeneration, and remodeling, together with the powerful profibrotic signaling they provide upon activation, via secreted serotonin (5-HT), transforming growth factor β (TGF-β), and platelet-derived growth factor (PDGF) [65,66]. While platelets themselves are important sources of TGF-β and PDGF, platelet-derived serotonin also has the potential to stimulate additional immune and connective tissue cells to secrete TGF-β [67,68].

In addition to the profibrotic platelet mediator and cytokines mentioned above, the platelet-specific gene signature derived in our study included five transcripts (that is, *CLU*, *PF4*, *PPBP*, *SPARC*, and *THBS1*) with potent extracellular matrix (ECM) deposition and fibrosis-promoting functions. It is important to note that *THBS1* (FC: 1.850), *SPARC* (FC: 1.500), and *PPBP* (FC: 1.428) were among the genes most differentially expressed in our platelet-specific gene signature (Table 3). *PF4* (platelet factor 4, also known as *CXCL4*) is an alpha granule content, a chemotactic factor, and a potent myofibroblast activator that stimulates excessive ECM deposition [69]. *PPBP* (pro-platelet basic protein, also known as *CXCL7*), another powerful chemokine, also functions as a platelet-derived growth factor that stimulates the synthesis of ECM hyaluronic acid and sulfated glycosaminoglycans [70]. *SPARC* (secreted protein acidic and cysteine rich) and *THBS1* (thrombospondin 1) are two ECM-associated platelet proteins with fibroblast-proliferating, collagen-producing, and ECM-depositing characteristics [71,72].

*MIR223* deserves special mention. *MIR223* was the only transcript in the platelet-specific gene signature with decreased expression (FC: −1.045) in the IgG4-RD patient group (Table 3). While platelets are capable of de novo protein synthesis, their miRNA content is also capable of regulating the translation process [73,74]. The *MIR223* transcript has been shown to be the most abundant miRNA molecule in platelets and one of the key targets of miR-223 is *P2RY12*, the ADP receptor in platelets [73]. This miRNA–mRNA target relationship appears to be responsible for the observed high on-treatment platelet reactivity with decreased platelet miR-223 expression [75]. The documented decreased expression level of *MIR223* in our study may have contributed to the “activated” phenotype of platelets.

Four recent articles reviewed the pathogenesis of IgG4-RD in detail [11,12,62,64]. These studies documented both innate and adaptive immune mechanisms in IgG4-RD’s pathogenesis with a possibility that innate immune abnormality might be preceding adaptive immunity [11,62]. While tissue infiltration with IgG4(+) plasma cells is a hallmark of the condition, abnormal T cell activity, especially CD4+ cytotoxic T lymphocyte and follicular T helper lymphocyte activity, has a pivotal role in IgG4-RD immunopathogenesis [11,12,62,64]. Indeed, most lymphocytes present in IgG4-RD tissues are shown to be T lymphocytes [11]. It was shown that IgG4-RD-associated follicular T helper cells were capable of inducing the differentiation and proliferation of B cells higher than their normal counterparts [11]. Interestingly, IL-4, which is a follicular T helper cytokine, induced class switch recombination resulting in IgG4 production [11]. Aberrant T helper type 2 cells have been put forth to promote tissue fibrosis through profibrotic cytokines (i.e., IL-4, IL-5, and IL-13) in IgG4-RD [11,62]. Also, in some IgG4-RD patients, an increase in CD4+CD25+ Treg cells was documented. These Treg cells were capable of IL-10 and TGF-B secretion, which contribute to IgG4 class switching and fibrosis, respectively [11]. Strong evidence for the central role of B cells in IgG4-RD comes from the robust response of the condition to B cell depletion therapies [12,62]. Despite this wealth of information, IgG4-RD pathogenesis is far from being clearly explained [12,62,64]. In particular, the precise mechanisms leading to tissue fibrosis are waiting for explanations [62]. Importantly, none of the abovementioned articles exploring the pathogenesis of IgG4-RD mentioned platelets in any context [11,12,62,64].

Another significant study was performed by Cai et al. [18]. Cai et al. performed a proteomic analysis to investigate the pathogenetic mechanisms of IgG4-RD and used two transcriptomic data sets (GSE66465 by Nakajima et al. and GSE40568 by Tsuboi et al.) from the GEO data repository for the validation of their findings [18]. In serum and tissue samples from IgG4-RD patients, Cai et al. documented several platelet-related terms (i.e., “platelet activation”, “platelet degranulation”, “platelet aggregation”, “blood coagulation”, “hemostasis”, and “coagulation”) to be enriched, among many other titles [18]. As authors, we believe that the concordance of the findings of Cai et al. which were reached using a different omics strategy in two different tissues (i.e., blood and submandibular glands) is strongly in support of our findings [18]. An observational study by Gutierrez et al. explored the epidemiology and risk factors of arterial and venous thrombotic events in patients with IgG4-RD and documented that arterial and venous thrombotic complications are common in patients with IgG4-RD [76]. It is noteworthy that Gutierrez et al. concluded in their study report, “mechanisms responsible for this over-risk and clinical benefit of a preventive platelet antiaggregant or anticoagulant treatment in high risk of thrombosis subgroups remain to be evaluated” [76].

As with any study, this study also has limitations to mention. The limited counts of IgG4-RD patients and control samples are presumably the most remarkable limitation of our study. This fact raises the need to plan future research gathering a large number of IgG4-RD patients and controls. Such large sample studies will also compensate for any potential heterogeneity in IgG4-RD immunopathogenesis with respect to its distinct disease subsets, as occurs in many other inflammatory conditions. Another limitation may arise from the design of the study. As mentioned before, we implemented a “secondary data analysis” approach for our study. Although this means that we were using the research data of other scientists which have already become available for public use and did not belong to our group, this study design is becoming more popular among researchers and surely also has certain advantages [77,78]. The use of high-throughput omics technologies resulted in an enormous increase in valuable omics data. Secondary analysis of these omics data has the advantage of saving time, money, and many other valuable resources [77]. Finally, the “in silico” nature of the design of our study and the lack of validation of our findings with carefully designed in vitro studies should be mentioned as other important limitations of the present study.

## 5. Conclusions

This preliminary study performed a secondary analysis of existing transcriptomic data on IgG4-RD, to find interesting clues for a potential platelet contribution to IgG4-RD immunopathogenesis. The 22-gene platelet-specific signature constructed from the DEG set of the study and the functional enrichment findings of the same 268-gene DEG set pointed to the presence of a clear platelet activation/degranulation process in IgG4-RD patients. Using today’s technologies that allow us (1) to purely isolate platelets, platelet-derived microvesicles and various leukocyte-platelet aggregates, (2) to perform single cell and bulk RNA sequencing, and (3) to thoroughly characterize proteomes, if the role of platelets in IgG4-RD pathogenesis is validated, future research should evaluate the therapeutic potentials of conventional and/or novel antiplatelet drugs in IgG4-RD. When planning transcriptomics and/or proteomics research with platelets in the spotlight, relatively low amounts of platelet mRNA, miRNA, and protein content should be taken into account. With carefully presented hypotheses and meticulously designed studies, secondary analysis of omics data sets has great potential to reveal new and valuable information.

## Figures and Tables

**Figure 1 medicina-61-00162-f001:**
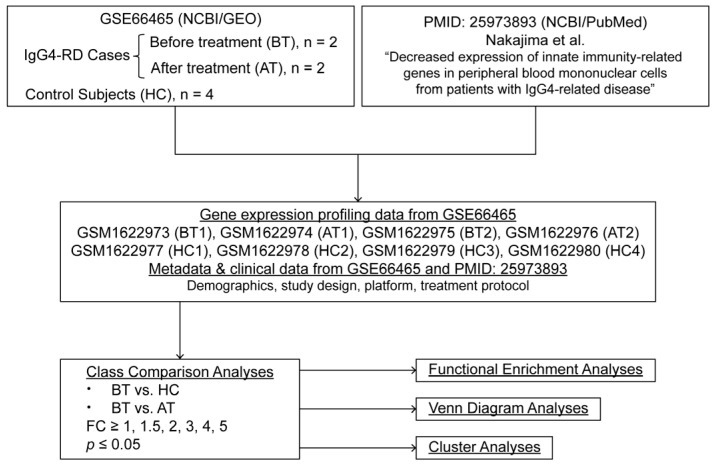
Flow diagram of the study. PMID: 25973893 is the article by Nakajima et al. [19].

**Figure 2 medicina-61-00162-f002:**
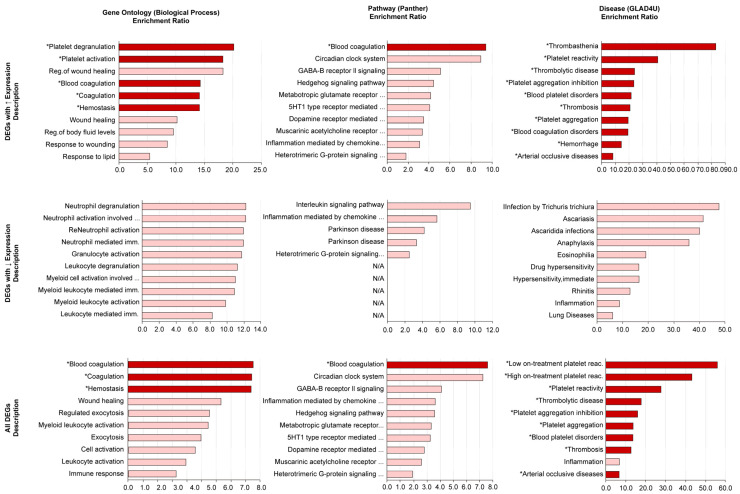
The graphical representations of the functional enrichment analysis results. While the columns represent the functional databases searched (from left to right, gene ontology, pathway, and disease), the rows represent the DEG sets used (from the top to the bottom, only the DEGs with increased expression levels, only the DEGs with decreased expression levels, and the combined DEG set). * Gene sets regarding platelets, platelet functions, and platelet activation.

**Figure 3 medicina-61-00162-f003:**
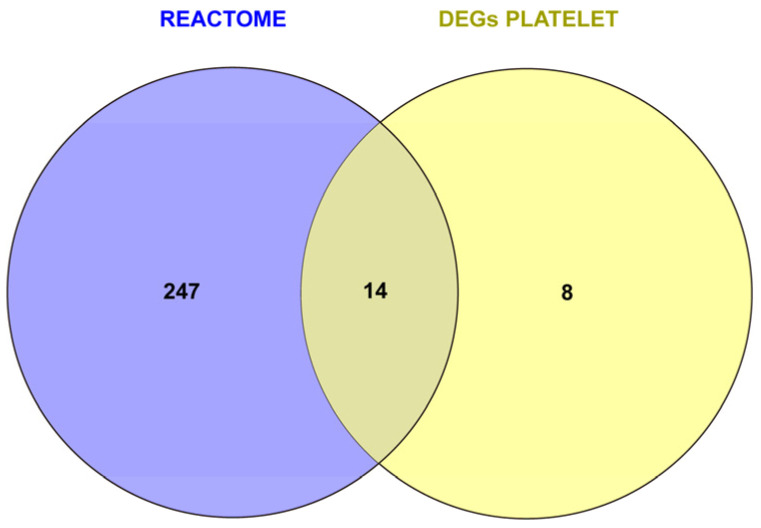
Graphic representation of the Venn diagram analysis of the platelet-specific gene signature (DEGs PLATELET) and the Reactome database’s “platelet activation, signaling, and aggregation” pathway (REACTOME). REACTOME ∩ DEGs PLATELET includes *CLEC1B*, *CLU*, *GP1BA*, *ITGA2B*, *ITGB3*, *MMRN1*, *MPL*, *P2RY12*, *PF4*, *PPBP*, *PROS1*, *SELP*, *SPARC*, and *THBS1*.

**Figure 4 medicina-61-00162-f004:**
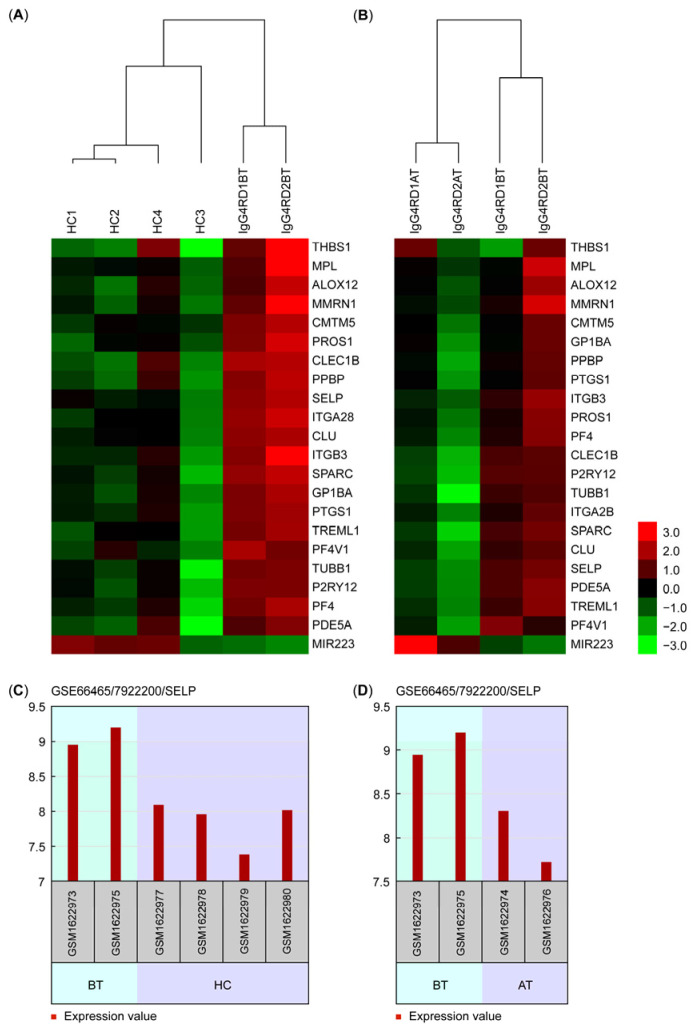
Dendrogram and heat map representations of the cluster analyses (panels (**A**,**B**), where rows represent genes and columns represent samples) and graphical presentations of the relative *SELP* gene expressions (panels (**C**,**D**)). (**A**) Dendrogram and heat map of the BT and HC clustering; (**B**) dendrogram and heat map of the BT and AT clustering; (**C**) relative *SELP* expression levels during the BT vs. HC class comparison; (**D**) relative *SELP* expression levels during the BT vs. AT class comparison.

**Table 1 medicina-61-00162-t001:** The results of the class comparison analyses of BT vs. HC and BT vs. AT.

		Number of Differentially Expressed Genes					
		BT vs. HC			BT vs. AT		
FC	*p*	Total	Increased ^1^	Decreased ^2^	Total	Increased ^1^	Decreased ^2^
≥1	≤0.05	268	204	64	230	107	123
≥1.5	≤0.05	77	49	28	61	13	48
≥2	≤0.05	25	13	12	33	3	30
≥3	≤0.05	5	-	5	14	-	14
≥4	≤0.05	1	-	1	7	-	7
≥5	≤0.05	-	-	-	3	-	3

^1^ Increased expression in the first class (e.g., BT) compared to the second class (e.g., HC). ^2^ Decreased expression in the first class (e.g., BT) compared to the second class (e.g., HC). AT, after treatment (glucocorticoid-treated IgG4-RD patients); BT, before treatment (treatment-naive IgG4-RD patients); FC, fold change; HC, healthy control.

**Table 2 medicina-61-00162-t002:** The top 30 most differentially expressed genes, obtained during the class comparison analyses of BT vs. HC and BT vs. AT.

Increased Expression ^1^			Decreased Expression ^2^		
Gene Symbol		FC	Gene Symbol		FC
**BT vs. HC**					
**1**	*HIST1H2BB*	2.69	**1**	*CLC*	−4.26
**2**	*NR4A2*	2.49	**2**	*MS4A3*	−3.34
**3**	*SNORD75*	2.42	**3**	*DEFA1B*	−3.21
**4**	*RGS1*	2.16	**4**	*LRRN3*	−2.70
**5**	*AREG*	2.04	**5**	*CXCR1*	−2.46
**6**	*SNORD28*	2.03	**6**	*HBB*	−2.31
**7**	*SNORD3D*	2.03	**7**	*CA1*	−2.24
**8**	*NR4A3*	1.96	**8**	*CXCR2*	−2.22
**9**	*RNU5E* *-* *1*	1.91	**9**	*MMP8*	−2.09
**10**	*SNORA80E*	1.90	**10**	*MME*	−2.07
**11**	*SNORD78*	1.85	**11**	*CRISP3*	−1.98
**12**	*THBS1*	1.85	**12**	*ALAS2*	−1.98
**13**	*SNORD50B*	1.73	**13**	*CPA3*	−1.95
**14**	*HIST1H3J*	1.73	**14**	*SLC25A37*	−1.82
**15**	*SNORA4*	1.72	**15**	*FCER1A*	−1.79
**BT vs. AT**					
**1**	*HIST1H2BB*	2.47	**1**	*DEFA1B*	−5.95
**2**	*IFI44L*	2.16	**2**	*MMP8*	−4.74
**3**	*SNORA80E*	2.09	**3**	*MS4A3*	−4.63
**4**	*TNFRSF17*	1.96	**4**	*DEFA4*	−4.37
**5**	*SNORD75*	1.64	**5**	*CEACAM8*	−4.13
**6**	*GNG11*	1.64	**6**	*CA1*	−3.70
**7**	*CD38*	1.59	**7**	*OLFM4*	−3.61
**8**	*HIST1H3J*	1.53	**8**	*CRISP3*	−3.56
**9**	*SNORD74*	1.50	**9**	*CEACAM6*	−3.46
**10**	*HIST2H2BF*	1.46	**10**	*AHSP*	−3.45
**11**	*IGKC*	1.46	**11**	*BPI*	−3.35
**12**	*SLC25A20*	1.40	**12**	*CD177*	−3.06
**13**	*SNORA61*	1.40	**13**	*CLC*	−2.90
**14**	*MZB1*	1.39	**14**	*ALAS2*	−2.78
**15**	*CAV1*	1.36	**15**	*CTSG*	−2.62

^1^ Increased expression in the first class (e.g., BT) compared to the second class (e.g., HC). ^2^ Decreased expression in the first class (e.g., BT) compared to the second class (e.g., HC). AT, after treatment (glucocorticoid-treated IgG4-RD patients); BT, before treatment (treatment naive IgG4-RD patients); FC, fold change; HC, healthy control.

**Table 3 medicina-61-00162-t003:** The list, FC and *p* values, and key platelet-related functions of the shared 22 genes that appear both in our DEG list and in the enriched platelet-specific terms’ gene lists ^1^.

Gene	Log FC	*p* Value	Function
*ALOX12*	1.000	8.36 × 10^−3^	Platelet activation and aggregation
*CLEC1B*	1.421	9.23 × 10^−4^	Platelet activation and aggregation, thromboinflammation
*CLU*	1.286	1.12 × 10^−3^	Platelet alpha granule content
*CMTM5*	1.007	2.19 × 10^−4^	Platelet reactivity, overexpression in platelets
*GP1BA*	1.164	3.33 × 10^−3^	Platelet receptor for von Willebrand factor (VWF), platelet adhesion
*ITGA2B*	1.402	3.88 × 10^−4^	Platelet receptor for fibrinogen (FI), platelet aggregation
*ITGB3*	1.475	3.84 × 10^−3^	Platelet receptor for fibrinogen (FI), platelet aggregation
*MIR223*	−1.045	1.17 × 10^−2^	Platelet aggregation, secretion, and reactivity
*MMRN1*	1.228	5.09 × 10^−3^	Platelet adhesion, platelet-derived FV carrier
*MPL*	1.013	1.04 × 10^−2^	Thrombopoietin receptor, platelet production
*P2RY12*	1.030	1.52 × 10^−2^	Platelet purinergic (ADP) receptor, platelet aggregation
*PDE5A*	1.072	5.65 × 10^−2^	Agonist stimulated platelet adhesion, aggregation, and secretion
*PF4*	1.265	1.89 × 10^−2^	Platelet alpha granule content, platelet aggregation and secretion
*PF4V1*	1.073	3.32 × 10^−3^	Platelet factor 4 (PF4) homolog, chemokine
*PPBP*	1.428	2.98 × 10^−3^	Platelet alpha granule content, growth factor (ECM synthesis), chemokine
*PROS1*	1.154	5.81 × 10^−4^	Platelet alpha granule content, platelet activation and aggregation
*PTGS1*	1.182	8.49 × 10^−3^	Platelet activation and aggregation, also known as COX1
*SELP*	1.202	5.08 × 10^−4^	Platelet alpha granule content, platelet activation and aggregation
*SPARC*	1.500	4.50 × 10^−3^	Platelet alpha granule content, platelet aggregation, ECM organization
*THBS1*	1.850	2.77 × 10^−2^	Platelet alpha granule content, platelet aggregation, ECM organization
*TREML1*	1.212	7.33 × 10^−3^	Platelet alpha granule content, platelet activation and aggregation
*TUBB1*	1.105	3.87 × 10^−2^	Platelet production, platelet aggregation and reactivity

^1^ Genes are presented in alphabetical order. ADP, adenosine diphosphate; COX1, cyclooxygenase 1; ECM, extracellular matrix; FV, Factor V; FC, fold change.

## Data Availability

All the bioinformatics data supporting the findings of this study are either presented within the manuscript or within the Appendix A accompanying the manuscript. The microarray expression data of the study by Nakajima et al. can be obtained through the Gene Expression Omnibus data repository with the accession number GSE66465 [19].

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
