# Peer review of "The Platelet-Specific Gene Signature in the Immunoglobulin G4-Related Disease Transcriptome"

_medicina, 2025, doi:10.3390/medicina61010162_

Round 1
Reviewer 1 Report
Comments and Suggestions for Authors
This publication is characterized by an excellent level of writing: a detailed introduction, many relevant references and necessary explanations in the course of the text. The study seems to be well-founded, the level of competence of the authors does not raise any questions. However, there are a number of minor shortcomings: the design of the figures in the article is poor, the font is small, the quality of the images is poor. The abstract is written atypically for MDPI journals, obviously the authors did not revise it when resubmitting it from another journal. Among the more significant flaws - there are many questions about the authors' use of other people's data to construct their study, which they themselves write about. The number of patients with IgG4-RD - 2 patients - is clearly insufficient for a reliable assessment, especially considering that IgG4-RD is a very heterogeneous disease with very variable manifestations. What criteria were used to select these patients is also not specified. Therefore, the results obtained are not credible.
Author Response
1
AUTHORS’ RESPONSES TO REVIEWERS’ COMMENTS
(medicina-3393526)
Reviewer 1
Dear Reviewer,
As the authors of the manuscript “The Platelet-Specific Gene Signature in the Immunoglobulin G4-Related Disease Transcriptome” (medicina-3393526), we express our sincere gratitude to you for your time and effort to review our manuscript and also for your objective and constructive criticism regarding our article. We are sure that, the revised and improved manuscript in line with your comments, will much more clearly and correctly present our preliminary findings to the related scientific community. On the special occasion of the New Year, we all wish you the best of everything, especially health and success in 2025.
We are presenting our detailed responses to your comments and the corresponding revisions of the manuscript below, also with highlighted text in our re-submitted file.
Questions for General Evaluation
•Does the introduction provide sufficient background and include all relevant references? (Reviewer’s Evaluation: Yes) [Thank you very much for your evaluation.]
•Is the research design appropriate? (Reviewer’s Evaluation: Must be improved) As you have commented and the authors of the manuscript totally agree, the two major limitations of the presented study are the limited number of patients & controls evaluated and its “in silico” secondary analysis design. Both of these important limitations were mentioned and potential solutions discussed in the final paragraph of the “Discussion” section (pages: 12-13, lines: 404-417). In the current situation, three authors of the study (A.K.O., C.S.O., and S.S.K.) are in preparation for an in vitro research project which will explore the potential contribution of platelets to IgG4-RD pathogenesis. As a relatively rare disorder and a significantly heterogenous disease with respect to organ involvement, the planned research will need time to recruit patients. Additionally, in order to emphasize the “in silico” nature of the design of the study and the lack of validation of our “in silico” findings, we revised the last sentence of the last paragraph of the “Discussion” section as (page: 13, lines: 417-419): “Finally, the “in silico” nature of the design of our study and the lack of validation of our findings with carefully designed in vitro studies should be mentioned as other important limitations of the present study.”
•Are the methods adequately described? (Reviewer’s Evaluation: Can be improved) In accordance with your comments regarding the “Methods” section of the manuscript, the authors thoroughly and carefully read this section with a critical eye.
The following revisions were made: (1) The statement was added (page: 3, lines: 96-97): “As the study involved an “in silico” secondary analysis of a publicly available transcriptomic data set on IgG4-RD, ethical approval was not sought.” (2) The sentence was added (page: 3, lines: 104-107): “The comprehensive diagnostic criteria for IgG4-RD (2011) by Umehara et al. involved two diagnostic criteria: (1) serum IgG4 concentration >135 mg/dL and (2) >40% of IgG (+) plasma cells being IgG4 (+) and >10 cells/high-power field of biopsy sample 20.” (3) With regard to the study population, the information was added (page: 3, lines: 107-111): “The two IgG4-RD patients were males with the ages of 66 (patient 1) and 63 (patient 2) 19. Four control subjects were also all males, ages ranging from 57 to 64 19. The tissue/organ involvement of the two patients were as salivary gland/duodenum/lymph node and salivary gland/bile duct/pancreas/prostate for the patient 1 and the patient 2, respectively 19.” (4) About the in vitro experiments of Nakajima et al., the following statement was added (page: 3, lines: 115-116): “The details regarding the RNA isolation and the microarray hybridization steps are presented in Nakajima et al.’s paper 19.”
•Are the results clearly presented? (Reviewer’s Evaluation: Must be improved) Just as you expressed in your comments, we, because of our limited experience with figure/graphic design, were not able to prepare striking and informative figures supplementing our article. At this point, we decided to receive professional help from the MDPI (Author Services) Figure Editing service. The necessary image files were uploaded and all the other additional procedures completed on December 29th 2024 (ID: 88871). Additionally, the authors decided to receive professional help for Layout Editing of the manuscript from the MDPI Author Services again, following their completion of the review process of the article according to the reviewers’ comments (January 2nd 2025, ID: 88965).
•Are the conclusions supported by the results? (Reviewer’s Evaluation: Must be improved) As the authors of the manuscript, we presented three main conclusions in the final “Conclusions” section. As you have stated in your Review Report and as the authors completely agree, because of its limitations pertaining to its design, this preliminary study should present its conclusions carefully and cautiously. Accordingly, we revised the first sentence of the “Conclusions” section as (page: 13, lines: 421-423): “This preliminary study performed a secondary analysis of an existing transcriptomic data on IgG4-RD, to find interesting clues for a potential platelet contribution to IgG4-RD immunopathogenesis.” Regarding the other two main conclusions presented in our article (i.e., (1) the relatively low amount of platelet mRNA and its importance when planning transcriptomics research exploring platelets and (2) the potential value of the secondary analysis of omics data sets to reveal new and valuable information
), the authors believe that, both the findings of the study and the presented literature are in support of these conclusions.
Point-by-point Response to Comments and Suggestions for Authors
Comment 1: “This publication is characterized by an excellent level of writing: a detailed introduction, many relevant references and necessary explanations in the course of the text. The study seems to be well-founded, the level of competence of the authors does not raise any questions.”
Response 1: Thank you very much for your positive comments.
Comment 2: “However, there are a number of minor shortcomings: the design of the figures in the article is poor, the font is small, the quality of the images is poor.”
Response 2: The authors totally agree with this comment and have submitted all the 4 figures of the manuscript to MDPI Author Service for Figure Editing (December 29th 2024, ID: 88871).
Comment 3: “The abstract is written atypically for MDPI journals, obviously the authors did not revise it when resubmitting it from another journal.”
Response 3: Dear Reviewer, As the authors of the manuscript, we did our best to prepare and upload an “Abstract” and a Main Text File to the submission system which is in accord with Medicina’s submission requirements, by (1) using the Microsoft Word Template provided by Medicina, (2) using exactly/no more than 300 words, and (3) using the abstract subheadings (Background and Objectives, Materials and Methods, Results, Conclusions) specifically mentioned in the template. Without any doubt, this is the authors first submission to an MDPI Journal and the authors are ready for any further corrections regarding the “Abstract” of their manuscript.
Comment 4: “There are many questions about the authors' use of other people's data to construct their study, which they themselves write about.”
Response 4: When the “secondary analysis” design of the study is compared to an “in vitro” transcriptomics study, it may surely be reported as a limitation of the present study. The authors agree with this comment and mentioned it a couple of times throughout the manuscript (page: 2, line: 87; page: 3, line: 96; page: 10, line: 277; page: 13, line: 410; page: 13, line: 421; page: 13, line: 445). Nonetheless, as Price et al. stated “The potentially greater value of these data lies in their secondary utilization as the deployment of data science and artificial intelligence in biology advances.” (What is the real value of omics data? Enhancing research outcomes and securing long-term data excellence, https://doi.org/10.1093/nar/gkae901), the authors also believe that there is a huge amount of valuable data hidden in the publicly open omics data sets. Four of the authors (A.K.O., C.S.O., S.S.K., and I.E.) have previously published another “secondary analysis” based research paper on Behcet syndrome immunopathogenesis (Behçet's: A Disease or a Syndrome? Answer from an Expression Profiling Study, https://doi.org/10.1371/journal.pone.0149052). As the amount of omics data present in the public domains is increasing every day, an increasing number of secondary analysis research papers are expected, while definitely keeping the ethical regulations concerning them. By clearly depicting the identity of the database, the date of accession, the accession number, and the authors (page: 3, lines: 119-120), we have fulfilled all the obligatory ethical prerequisites for a secondary analysis research paper. Additionally, our “Acknowledgments” stated (page: 13, lines: 452-453), “The authors express their sincere gratitude to all the scientists who share their valuable research data with their colleagues for the benefit of humanity.”
Comment 5: “The number of patients with IgG4-RD - 2 patients - is clearly insufficient for a reliable assessment, especially considering that IgG4-RD is a very heterogeneous disease with very variable manifestations.”
Response 5: Authors clearly agree with this comment and declared this limitation in the manuscript (page: 12, lines: 404-406; page: 12, lines: 407-409), presented the study as a “preliminary” study (page: 2, line: 92; page: 13, line: 421), and stated the relevant conclusion with carefully selected words (page: 2, lines: 92-94; page: 13, lines: 421-423). As stated above, three of the authors of the manuscript are working on an in vitro research project focusing on the potential contribution of platelets to IgG4-RD pathogenesis.
Comment 6: “What criteria were used to select these patients is also not specified.”
Response 6: We express our sincere thanks to you on your pointing of this important unintentional omission. Accordingly, we have added this important information as (page: 3, lines: 102-107): “Two patients with IgG4-RD diagnosed according to the comprehensive diagnostic criteria and 4 healthy control subjects were enrolled in the gene expression profiling study by Nakajima et al. (GEO accession GSE66465) [19,20]. The comprehensive diagnostic criteria for IgG4-RD (2011) by Umehara et al. involved two diagnostic criteria: (1) serum IgG4 concentration >135 mg/dL and (2) >40% of IgG (+) plasma cells being IgG4 (+) and >10 cells/high-power field of biopsy sample [20].”
Response to Comments on the Quality of English Language
Reviewer’s Evaluation: The quality of English does not limit my understanding of the research.
[Thank you very much for your evaluation.] The legend of the Figure 2 was revised and expanded (page: 7, lines: 218-220) “While the columns represent the functional databases searched (from left to right, Gene Ontology, Pathway, and Disease), the rows represent the DEG sets used (from the top to the bottom, Only the DEGs with increased expression levels,
Only the DEGs with decreased expression levels, and The combined DEG set). * Gene sets regarding platelets, platelet functions, and platelet activation.”
Reviewer 2 Report
Comments and Suggestions for Authors
Dear Authors,
your secondary analysis of publicly available omics data provided very interesting results on the role of platelets in G4-related disease pathogenesis. I have a couple of comments.
1. What is the frequency of G4-related disease? Is it age-related? Please, add this information to the Introduction.
2. You've justfully written that small sample size is one of the limitation of your study. However, some main information about patients and healthy donors is needed. Please, provide their ages and genders at least.
Author Response
Please see the attachment.
Reviewer 2
Dear Reviewer,
On behalf of the authors of the manuscript “The Platelet-Specific Gene Signature in
the Immunoglobulin G4-Related Disease Transcriptome” (medicina-3393526), I do
express my sincere gratitude to you for your precious time and effort spent to review
our manuscript. As the authors we are sure that, the revised and therefore improved
manuscript in line with your objective and constructive criticism, will much more clearly
and correctly present the preliminary findings of our study to the scientific community.
Also, on the special occasion of the New Year, we all wish you the best of everything,
especially health and success in 2025.
We are presenting our responses to your comments and the corresponding revisions
of the manuscript below, also with highlighted text in our re-submitted file.
Questions for General Evaluation
• Does the introduction provide sufficient background and include all relevant
references? (Reviewer’s Evaluation: Yes) [Thank you very much for your
evaluation.]
• Is the research design appropriate? (Reviewer’s Evaluation: Yes) Thank
you very much for your evaluation.
• Are the methods adequately described? (Reviewer’s Evaluation: Yes) Thank
you very much for your evaluation.
• Are the results clearly presented? (Reviewer’s Evaluation: Yes) Thank you
very much for your evaluation.
• Are the conclusions supported by the results? (Reviewer’s Evaluation: Yes)
Thank you very much for your evaluation.
Point-by-point Response to Comments and Suggestions for Authors
Comment 1: “… your secondary analysis of publicly available omics data provided
very interesting results on the role of platelets in G4-related disease pathogenesis.”
Response 1: Dear Reviewer, Thank you very much for your positive comment.
Comment 2: “What is the frequency of G4-related disease? Is it age-related? Please,
add this information to the Introduction.”
Response 2: The authors totally agree with your comment and are grateful to you for
your critical reminding of this important point. Accordingly, we have added the following
passage to the “Introduction” section (page: 2, lines: 66-73): “A recent review
reports that IgG4-RD tends to occur during the fifth to seventh decades of life,
although both pediatric and elderly patients are also observed 13. With regard
to gender, while most IgG4-RD studies document a male predominance, this
gender difference shows variation with the tissues/organs involved (i.e.,
pancreatobiliary and retroperitoneal involvement more common among males
and disease confined to head & neck region more common among females) 13.
Although population specific measures of incidence and prevalence are scarce
in the literature, the estimated incidence was reported to be around 0.78-1.39
per 100,000 person-years in the USA 14.” This passage included two new
references.
Comment 3: “You've justfully written that small sample size is one of the limitation of
your study. However, some main information about patients and healthy donors is
needed. Please, provide their ages and genders at least.”
Response 3: Again, all the authors agree with your comment and express their
sincere gratitude for your crucial remark. With regard to the study population, the
following information was added (page: 3, lines: 107-111): “The two IgG4-RD patients
were males with the ages of 66 (patient 1) and 63 (patient 2) 19. Four control
subjects were also all males, ages ranging from 57 to 64 19. The tissue/organ
involvement of the two patients were as salivary gland/duodenum/lymph node
and salivary gland/bile duct/pancreas/prostate for the patient 1 and the patient
2, respectively 19.”
Response to Comments on the Quality of English Language
Reviewer’s Evaluation: The quality of English does not limit my understanding of the
research.
[Thank you very much for your evaluation.]
Reviewer 3 Report
Comments and Suggestions for Authors
In this manuscript, the authors performed a secondary analysis of existing transcriptome data on IgG4-RD to investigate the involvement of platelets in the pathogenesis of IgG4-RD.
I have a few comments on the paper.
Major points:
1) The main limitation of this study is the low sample size. Very small samples undermine the validity of the study.
2) Moreover, the research work would be more valuable if the Authors included the results of their own study group.
3) I further think that the Authors have to better emphasize the novelty of the study in comparison with previously published papers thus highlighting potential prognostic/clinical implications.
4) The Authors have unclearly specified which results are from Nakajima et al. and which are their own.
5) The “Results” section should be extended to better describe the analysis performed. The results should be described point by point.
Minor points:
1) The manustript lacks conclusions from the conducted studies.
2) In the manuscript, the tables 2 and 3 should be put as near as possible to the place where they are first referred to. Likewise, Figure 2 should be placed near the point of first mention in the text of the manuscript.
Round 2
Reviewer 1 Report
Comments and Suggestions for Authors
The authors have considered all comments, and the article can be published in its current form.
Reviewer 3 Report
Comments and Suggestions for Authors
I would like to thank the authors for taking my comments and suggestions into consideration and implementing them into the revised manuscript.